# Asymptotically fault-tolerant programmable photonics

Ryan Hamerly [1,2] ✉, Saumil Bandyopadhyay [1] & Dirk Englund [1]

Component errors limit the scaling of programmable coherent photonic circuits. These errors arise because the standard tunable photonic coupler—the Mach-Zehnder interferometer (MZI)—cannot be perfectly programmed to the cross state. Here, we introduce two modified circuit architectures that overcome this limitation: (1) a 3-splitter MZI mesh for generic errors, and (2) a broadband MZI+Crossing design for correlated errors. Because these designs allow for perfect realization of the cross state, the matrix fidelity no longer degrades with increased mesh size, allowing scaling to arbitrarily large meshes. The proposed architectures support progressive self-configuration, are more compact than previous MZI-doubling schemes, and do not require additional phase shifters. This removes a key limitation to the development of very-large-scale programmable photonic circuits.

Large-scale programmable photonic circuits are opening up radical new possibilities for optics. Of key importance in many devices is the universal multiport interferometer, which functions as an $N \times N$ reconfigurable feedforward linear circuit. This device, typically constructed with a compact mesh of Mach–Zehnder interferometers (MZIs, Fig. 1a, b)[1,2], is widely employed in applications ranging from spatially multiplexed optical communications to machine learning and quantum computing[3–7]. Sadly, component errors (Fig. 1c) are a critical factor limiting the size of such circuits. Since the circuit depth of MZI meshes scales as $O(N)$, the effect of errors grows with mesh size, meaning that, in practice, even modestly sized circuits cannot be programmed to high accuracy. Motivated by this challenge, a large body of recent work has focused on "correcting" hardware errors by global optimization[8–10], self-configuration[11–18], or local correction[19,20]. For conventional MZI meshes, correction reduces errors by a quadratic factor[16,19]; however, the effect of errors still grows with mesh size and poses a fundamental limit to the scaling of these circuits.

To overcome this limit, various alternative mesh architectures have been proposed. Non-compact structures such as binary trees avoid the extreme splitting-ratio requirements[21,22], but suffer from large chip area and the need for many crossings. A complementary approach is to stick to conventional geometries[1,2], but insert redundant MZIs to realize the full range of splitting ratios even in imperfect hardware[23–25]. This solves the scaling problem, but at the cost of a 1.5–2× increase in the number of splitters and phase shifters. The resulting effects on chip area (particularly on emerging high-speed platforms where phase shifters have a large footprint[26,27]), waveguide length (which affects insertion loss and latency[28]), and electronic complexity (number of pads, traces, DACs/drivers, etc.) make this option unappealing.

In this paper, we propose two mesh architectures that achieve the same perfect scaling without significant added complexity: a 3-splitter MZI that corrects all hardware errors (Fig. 1d) and an MZI+crossing design that only corrects correlated errors, but has the added advantage of broader bandwidth (Fig. 1e). These designs take up significantly less chip area than the "perfect" redundant MZIs[23,24], and do not require additional phase shifters. Moreover, the proposed architectures support progressive self-configuration[16,17], allowing for error correction even when the hardware errors are unknown. This work will enable the development of freely scalable, broadband, and compact linear photonic circuits.

This paper is structured as follows: first we introduce the formalism of error correction in MZI meshes, focusing on the self-configuration approach. Splitting ratios are visualized as points on the Riemann sphere, where forbidden regions emerge as a result of hardware imperfections; these regions are centered at the poles (bar- and cross-state), where the probability density is at a maximum. To avoid this unfortunate coincidence, our architectures "rotate" the Riemann sphere to move the forbidden regions away from this peak, so that a larger fraction of MZIs are perfectly realized. Based on this

[1]Research Laboratory of Electronics, MIT, 50 Vassar Street, Cambridge, MA 02139, USA. [2]NTT Research Inc., Physics and Informatics Laboratories, 940 Stewart Drive, Sunnyvale, CA 94085, USA. ✉e-mail: rhamerly@mit.edu

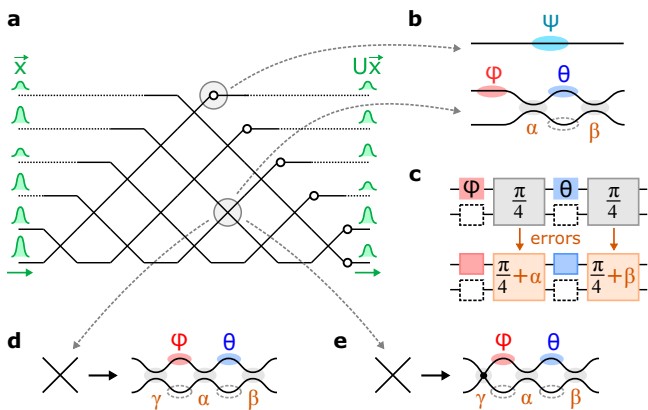

**Fig. 1 | Multiport interferometers with imperfect components. a** Universal $6 \times 6$ circuit realized by a triangular (Reck[1]) mesh. **b** Constituent components of the mesh include phase shifters ($\psi$) and programmable MZI couplers ($\theta, \phi$). **c** Fabrication imperfections lead to splitting-ratio errors $\alpha, \beta$. **d, e** Alternative error-resilient coupler designs proposed in this paper: **d** 3-splitter MZI and **e** MZI+crossing.

concept, we introduce the 3-splitter MZI, which can correct arbitrary errors by rotating the forbidden regions to the equator. Using a benchmark optical neural network, we show that this modified MZI mesh is >3× more robust to hardware errors, enabling accurate inference in a regime where standard interferometric circuits struggle. Finally, we introduce the MZI+crossing, which flips the poles of the Riemann sphere. While this design is only robust against correlated errors, it has the added advantage of broader intrinsic bandwidth. For both architectures, we compare the matrix fidelity to the standard MZI to demonstrate the scaling advantage of both schemes.

## Results

### Error correction formalism

To correctly configure an MZI mesh in the presence of errors, one uses a nulling method based on physical measurements[16,17]. Figure 2a illustrates the case of the triangular mesh[1], where the procedure is more straightforward. The transfer matrix for this system is a product of a phase screen $D$ and a sequence of $2 \times 2$ unitaries $W$:

$$U = D \underbrace{\prod_{mn} T_{mn}}_{W} \qquad (1)$$

where $T_{mn}$ is the $n$th MZI of the $m$th rising diagonal.

We program the mesh using a sequence of steps (Givens rotations), which build up $W$ in order to diagonalize the "target" matrix $X = UW^{\dagger}$. For every step, we add one $2 \times 2$ unitary to $W$, performing the update $W \rightarrow T_{mn}W$, which right-multiplies the target matrix $X \rightarrow XT_{mn}^{\dagger}$ (Fig. 2b). The $2 \times 2$ unitary $T_{mn}$ is chosen to null a specific matrix element $v \rightarrow 0$ (shaded green in Fig. 2b), which is equivalent to the equation (indices $m, n$ suppressed for notational simplicity):

$$[u\ v]T^{\dagger} = [* \ 0] \iff \frac{T_{11}}{T_{12}} = \frac{u}{v} \qquad (2)$$

This condition is visualized in Fig. 2c. In hardware, nulling of the $(i, j)$ element of $X$ is implemented by inputting the field $w_j^*$ ($j$th column of $W^{\dagger}$) and adjusting the MZI parameters ($\theta, \phi$) to zero the output power at the $i$th port. If all nulling steps are performed exactly, the mesh will perfectly realize the target matrix $U$ (see Methods and Supplementary Section 1 for details).

Mathematically, nulling involves setting the (complex-valued) splitting ratio $s \equiv T_{11}/T_{12} = -(T_{22}/T_{21})^*$ of the physical MZI to match the target value $\hat{s} \equiv u/v$ required for diagonalization. In many cases,

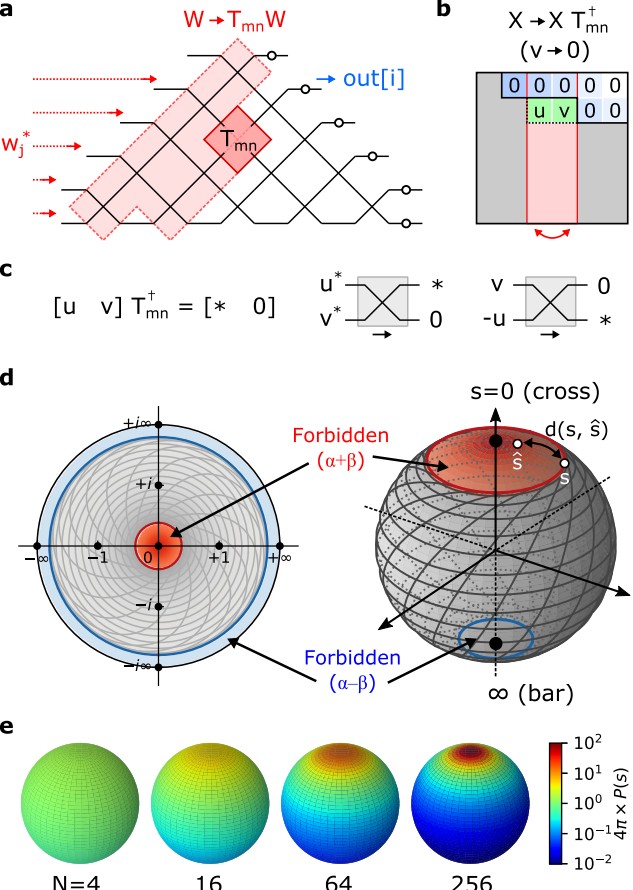

**Fig. 2 | Nulling method of self-configuration. a** Configuring MZI $T_{mn}$ updates matrix $W$. **b** Corresponding nulling update to $X = UW^{\dagger}$, which is **c** equivalent to zeroing an output of $T_{mn}$ given a fixed input. **d** Allowed range of $s = T_{11}/T_{12} \in \mathbb{C}$, showing the forbidden regions centered at $s = 0$ and $s = \infty$ that arise from hardware imperfections ($\alpha = 0.23$, $\beta = 0.07$ chosen for illustrative purposes). Contours for $(\theta, \phi)$ are plotted in the accessible region (gray). **e** Probability density $P(s)$ plotted on the Riemann sphere for meshes of size $N = 4$, 16, 64, and 256.

this is not possible, because the range of admissible splitting ratios $\tan|\alpha + \beta| \le |s| \le \cot|\alpha - \beta|$ is restricted in the presence of hardware imperfections, namely the splitting-angle errors for the 50:50 couplers in a real MZI ($\alpha, \beta$ in Fig. 1c). Owing to these imperfections, forbidden regions emerge for small and large $s$ where perfect nulling is impossible (Fig. 2d). It is also instructive to view this chart on the Riemann sphere, which shows that these forbidden regions are centered around the poles (Fig. 2b), highlighting the well-known fact that imperfect MZIs generally have finite extinction ratio and cannot realize a perfect cross ($s = 0$) or bar ($s = \infty$) state.

If in a given nulling step $\hat{s}$ falls within the forbidden region, nulling is imperfect, and an off-diagonal residual prevents perfect diagonalization of the matrix, leading to an "uncorrectable" error. This residual is proportional to $d(s, \hat{s})$, the Euclidean distance on the Riemann sphere between the target ratio and the closest realizable $s$. The overall error is the quadrature sum of all such residuals.

For linear photonic circuits, two important fidelity figures of merit are (1) the coverage $\mathcal{C}$, i.e., the probability that a matrix is realized exactly, and (2) the normalized matrix error $\mathcal{E} = \langle|\Delta U|_{\text{rms}}\rangle / \sqrt{N}$, a scaled Frobenius norm which is approximately equal to the average relative error for a given matrix element. $\mathcal{C}$ and $\mathcal{E}$ depend on the error model and the distribution of target matrices. Here, consistent with prior work[16,17,19,29], we sample target matrices randomly over the Haar measure[30,31] and consider an uncorrelated Gaussian error model $\langle\alpha\rangle_{\text{rms}} = \langle\beta\rangle_{\text{rms}} = \sigma$.

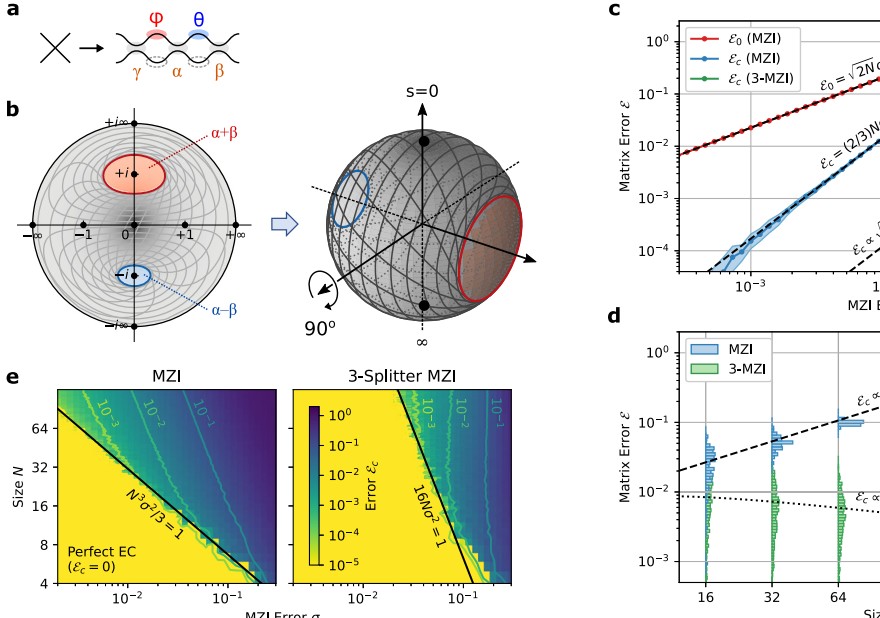

**Fig. 3 | 3-splitter MZI design and simulated performance. a** Schematic of 3-MZI. **b** Splitter Möbius transformation on $s \in \mathbb{C}$, which pushes the forbidden regions away from $s = \{0, \infty\}$, corresponding to a Riemann sphere rotation. **c** Dependence of matrix error $\mathcal{E}_0, \mathcal{E}_c$ on the splitter variation $\sigma$, contrasting the standard and 3-splitter MZIs (fixed mesh size $N = 256$). **d** Scaling of corrected error $\mathcal{E}_c$ with mesh size $N$, showing the qualitative scaling difference between MZI and 3-MZI (fixed splitter variation $\sigma = 0.05$). **e** Corrected error $\mathcal{E}_c$ as function of both $\sigma$ and $N$. The sudden onset of "perfect" hardware error correction ($\mathcal{E}_c = 0$) occurs when the coverage approaches unity ($\mathcal{C} \approx 1$).

Analytic expressions for $\mathcal{E}$ and $\mathcal{C}$ are derived in the Methods, which we summarize here. If a mesh is straightforwardly programmed without taking any account of the imperfections ("uncorrected" error), the normalized error is $\mathcal{E}_0 = \sqrt{2N}\sigma$[16,19]. The coverage $\mathcal{C} = e^{-N^3\sigma^2/3}$ (Eq. (16)) decreases sufficiently fast that even moderately sized meshes have vanishingly small coverage, and error correction is generally imperfect. In this case, the residual "corrected" error $\mathcal{E}_c = (2/3)N\sigma^2$ (Eq. (19)) is the more relevant metric. Since $\mathcal{E}_c \propto (\mathcal{E}_0)^2$, self-configuration correction affords a quadratic suppression of errors, which is a significant advantage when errors are below a threshold. However, for sufficiently large meshes $N \gtrsim 1/\sigma^2$, error correction will be ineffective and the mesh cannot realize most matrices at high fidelity. Thus, even with error correction, hardware imperfections set a fundamental scaling limit for standard MZI meshes.

## Asymptotically perfect photonic circuits

The main challenge limiting error correction here is that the forbidden regions overlap with the peak of the probability distribution, which clusters tightly around the cross state $s = 0$ (Fig. 2e)[29]. This clustering happens because light must propagate all the way down a mesh's diagonals to realize generic unitaries; the forbidden regions disrupt this ballistic transport leading to clipping of off-diagonal matrix elements[10]. Adding redundant components (MZI doubling) solves this problem by eliminating the forbidden regions altogether[23,24], but at the cost of added optical and electrical complexity. Here, we take the alternative approach of displacing the forbidden regions away from the cross state. This can be performed by placing a third splitter at the input of the MZI, as shown in Fig. 3a. The extra splitter performs a Möbius transformation $s \to (s + i \tan \eta)/(1 + is \tan \eta)$, which for a 50:50 splitting ratio ($\eta = \pi/4$) maps the bar and cross states states to $s = \pm i$ (Fig. 3b). This can be visualized as a 90° rotation on the Riemann sphere, which pushes the forbidden regions to the equator, while the probability density is still concentrated at the poles (small errors $\gamma$ in the third splitter perturb this rotation angle slightly, but this does not change the structure of the forbidden regions and has little effect on the error correction).

This "3-splitter MZI" (3-MZI) can therefore access the complete range of splitting-ratio magnitudes $|s| \in [0, \infty)$, and can thus function as a high-contrast optical switch[24,32]. However, forbidden regions are still present for the 3-MZI, which implies that for some configurations, the relative phase of the splitter arg($s$) is constrained by hardware errors (unlike the MZI-doubled "perfect" couplers of refs. [23–25], which cure this defect with redundant phase shifters). However, from the distributions in Fig. 2e, for large meshes $\hat{s}$ will fall into the 3-MZI's forbidden regions only rarely. The normalized matrix error, calculated in the Methods (Eq. (22)), takes the following form:

$$\mathcal{E}_c \approx 8\sigma^2 \left[ 2\frac{\log(N) - 1.366}{N} \right]^{1/2} \tag{3}$$

In Fig. 3c, d, we numerically simulate self-configuration on imperfect meshes using the MESHES package (see Methods and Supplementary code); the realized $\mathcal{E}_c$ shows good agreement with Eq. (3). For large meshes $N \gtrsim 64$, the matrix error is approximately 1–2 orders of magnitude lower with the 3-MZI. Moreover, the 3-MZI exhibits more favorable error scaling, with the error remarkably decreasing with mesh size as $\mathcal{E}_c \propto \sqrt{\log(N)/N}$. This leads to asymptotically fault-tolerant hardware error correction: in the limit $N \to \infty$, matrices can be programmed perfectly.

This non-intuitive effect arises from the fact that, under the Haar measure, only a small fraction of MZIs have significant probability density near $s = \pm i$, where the forbidden regions are centered[29]. This probability decreases exponentially with the distance from the triangle's base (see Methods for details). Therefore, although the mesh has $N(N-1)/2$ MZIs, only $O(N)$ contribute significantly to the matrix error under self-configuration. A naïve estimate assuming uncorrelated errors would give $|\Delta U| \propto \sqrt{N}\sigma^2$, which would lead to a constant $\mathcal{E}_c$. However, during the self-configuration process, subsequent MZIs can partially correct for errors in earlier MZIs that cannot be properly configured; the end result is to reduce the overall error of each MZI by a factor proportional to $\sqrt{\log(N)/N}$ (see Methods), yielding the result Eq. (3).

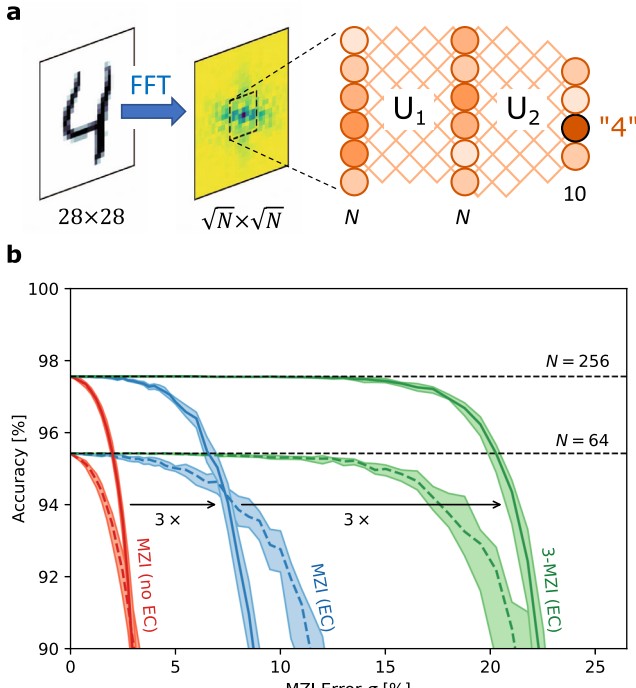

**Fig. 4 | Effect of hardware errors on DNN inference. a** Benchmark neural network consisting of FFT preprocessing, windowing, and two DNN layers, where the linear connections $U_1$ and $U_2$ are realized with MZI meshes[17,39]. **b** Inference accuracy as a function of MZI error.

A second benefit to the 3-MZI is its higher threshold for perfect error correction. One obtains this threshold by computing the coverage $\mathcal{C} = e^{-16N\sigma^2}$ (see Methods, Eq. (20)). This is much larger than the coverage of the regular MZI mesh, and the threshold scales as $\sigma_{th} \propto N^{-1/2}$, in contrast to the $N^{-3/2}$ dependence seen for the conventional mesh. Consequently, errors are perfectly correctable under a much broader range of circumstances, as shown in Fig. 3e.

### Error-resilient optical neural networks

To highlight the significance of this error reduction, consider as a concrete example deep neural network (DNN) inference on coherent optical hardware. A DNN is a sequence of layers, consisting of linear synaptic connections and nonlinear neuron activations. An emerging application of photonics seeks to use optical interference to accelerate this process, encoding neuron activations in coherent optical amplitudes, while a programmable MZI mesh implements the synaptic weights and activations are performed with an all-optical or electo-optic nonlinearity[5]. Scaling remains the major challenge to constructing practical optical neural networks, as large mesh sizes ($N > 100$) are required to achieve a significant advantages over electronic hardware, and such large meshes are especially susceptible to fabrication errors. A recent numerical study showed that even with state-of-the-art process tolerances, hardware errors can significantly degrade DNN inference accuracy[33], a difficulty that has spurred investigations into alternatives to the MZI mesh, which all have their own limitations[34–37].

Figure 4a depicts a benchmark neural network. Here, $28 \times 28$ images from the MNIST digit dataset[38] are preprocessed by a Fourier transform and cropped to a window of size $\sqrt{N} \times \sqrt{N}$, which forms the input to a two-layer unitary DNN. The DNN can be implemented optically with rectangular MZI meshes for synaptic weighting[2] and electro-optic nonlinearities for the activation (see refs. 17,39 for details). Models with inner-layer sizes $N = 64$ and $N = 256$ are pretrained using the NEUROPHOX package[40], and inference accuracy is

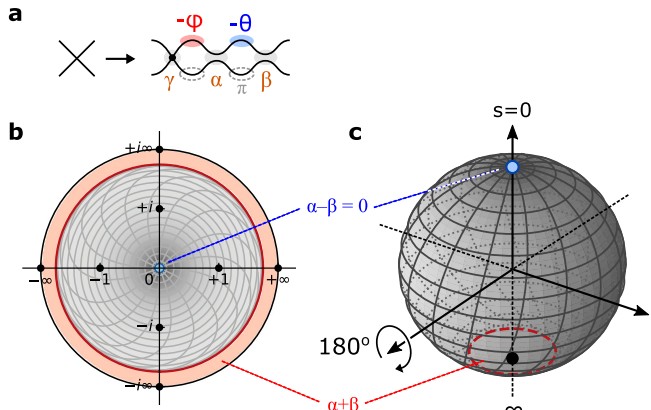

**Fig. 5 | MZI+crossing architecture. a** Schematic of MZI+X. **b** Effect of the crossing is to flip the $s = 0$ and $s = \infty$ forbidden regions. For correlated errors, the forbidden region around $s = 0$ disappears. **c** Riemann sphere projection.

subsequently simulated on imperfect meshes with Gaussian splitter errors to calculate the classification accuracy.

This accuracy is plotted in Fig. 4b for three cases: straightforwardly programming an MZI mesh without error correction, with error correction, and with the modified 3-MZI architecture. Even for small device errors $\sigma = 1–2\%$, which is considered state-of-the-art for directional couplers in highly controlled fabrication processes[41], hardware errors significantly degrade the model's inference accuracy relative to its canonical value ($\sigma = 0$). For small $\sigma$, this is recovered using error correction[17,19]. However, many broadband coupler designs[42–47] trade bandwidth for fabrication sensitivity and are in practice very sensitive to process variations, meaning larger splitter errors $\sigma \gtrsim 5\%$ are common. In this moderate-error regime, error correction alone is not sufficient and the network shows reduced accuracy, a problem that becomes more pronounced as the size $N$ increases. Moving to the 3-MZI architecture overcomes this limitation, enabling effectively error-free inference (relative to the canonical model) even out to very large splitter errors $\sigma \approx 10–15\%$, far beyond what is likely to be encountered in practice.

### Broadband mesh for correlated errors

For generic, uncorrelated component errors the 3-splitter MZI is well-suited. However, since the correlation lengths of process variations tend to be larger than a single MZI[48], errors are correlated in practice. This is especially true for broadband couplers based on multimode interference (MMI)[42,43], subwavelength gratings[44,45], and asymmetric designs[46,47], all of which are highly dependent on the device geometry, which can vary slightly from run to run. Moreover, even with perfect 50:50 couplers, the splitting ratios are still wavelength-dependent. Operating the mesh away from its design wavelength leads to correlated device errors, so sensitivity to these errors is closely tied to the operational bandwidth of the device.

Consider the case of a constant offset $\mu$ for all splitting ratios: $\alpha = \beta = \mu$. In a standard MZI, the bar-state forbidden region (around $s = \infty$) disappears since $|\alpha - \beta| = 0$, while the cross-state region (around $s = 0$, the peak of the probability distribution) remains in place (Fig. 2). This is consistent with the common observation that the extinction ratio in an MZI is much higher in the cross port than in the bar port. The optimal error reduction strategy, illustrated in Fig. 5a, was previously proposed in the context of broadband optical switching: place a waveguide crossing before the MZI[49]. The added crossing performs the Möbius transformation $s \to 1/s$, rotating the Riemann sphere by 180° to move the forbidden region to the minimum of the probability distribution (Fig. 5b, c).

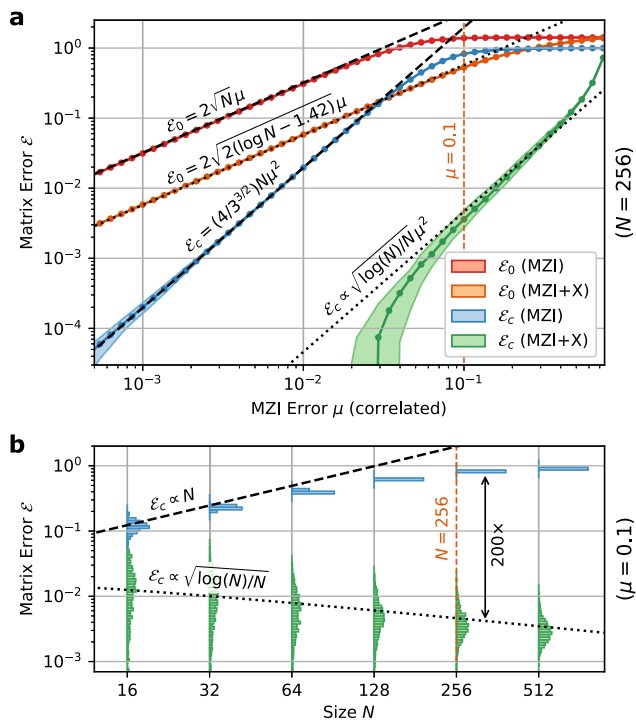

**Fig. 6 | Advantages of MZI+crossing architecture for correlated component errors. a** Dependence of matrix error $\mathcal{E}_0, \mathcal{E}_c$ on splitter error $\mu$ (fixed $N = 256$). **b** Dependence of $\mathcal{E}_c$ on mesh size $N$ (fixed $\mu = 0.1$).

As before, we can calculate the coverage and matrix error of this "MZI+crossing" (MZI+X) mesh by performing the nulling procedure on target unitaries, obtaining $\mathcal{C}$ from the probabilities that splitting ratios fall within the forbidden regions, and $\mathcal{E}_c$ from the residuals arising from imperfect diagonalization. In this case, there is only one forbidden region, centered at $s = \infty$. The calculation is worked out in the Methods. For the normalized error, we find (Eq. (27)):

$$\mathcal{E}_c = 4\mu^2 \left[ \frac{2}{3} \frac{\log(N) - 0.423}{N} \right]^{1/2} \quad (4)$$

This is plotted in Fig. 6. Like the 3-MZI design, this metric scales as $\mathcal{E}_c \propto \sqrt{\log(N)/N}\mu^2$, in contrast to the trend $\mathcal{E}_c = (4/3^{3/2})N\mu^2$ calculated for the standard MZI under correlated errors. The coverage also increases (Eq. (25)), so that the threshold for perfect correction likewise scales as $\mu_{\text{th}} \propto N^{-1/2}$, in contrast to the $\mu_{\text{th}} \propto N^{-3/2}$ dependence seen in the standard mesh.

Ultimately, the scalability of the MZI+X architecture is limited by differential errors $|\alpha - \beta|$ that arise from local fluctuations in waveguide dimensions. The effect of such errors is analyzed in Supplementary Section 2. For typical photonic process variations, $|\alpha - \beta| \ll \mu$ and differential errors are insignificant for mesh sizes up to at least $N = 512$.

As an added bonus, the MZI+X design also reduces the effect of errors in the absence of correction. To see how, we can make an analogy to Bloch-sphere rotations. The transfer matrix of a standard MZI is (up to a phase factor) the product of four rotations:

$$T(\theta, \phi) \propto R_x\left(\frac{\pi}{4} + \mu\right) R_z(\theta) R_x\left(\frac{\pi}{4} + \mu\right) R_z(\phi) \quad (5)$$

where $R_k(\eta) = e^{i\sigma_k \eta}$ is a Pauli rotation and $\sigma_k$ is a Pauli matrix. For the cross state ($\theta = 0$), the errors $\mu$ add up constructively, while for the bar state ($\theta = \pi$), they cancel out (the latter is a simple example of

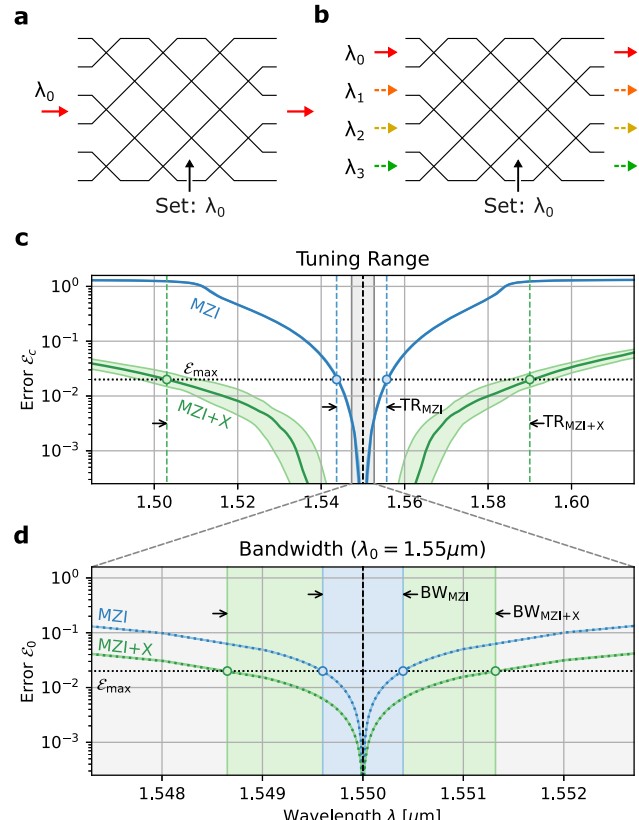

**Fig. 7 | Tuning range and bandwidth for MZI+X and standard MZI mesh, $N = 64$. a**, **b** Contrast between single- and multi-wavelength operation, which are limited by tuning range and bandwidth, respectively. **c** Plot of $\mathcal{E}_c(\lambda)$, which dictates the tuning range for a target matrix error $\mathcal{E}_{\text{max}}$. **d** Corresponding plot of $\mathcal{E}_0(\lambda)$, which dictates the bandwidth. Platform: $500 \times 220$ nm Si:SiO$_2$ directional coupler with 200 nm gap, $d\mu/d\lambda \approx 3.27/\mu$m.

dynamical decoupling of spins using a pulse sequence). Most crossings in large meshes are close to the cross state, which leads to constructive addition of the errors in the standard MZI mesh. However, for the MZI+X, the input ports of each MZI are exchanged, so the physical MZIs are close to the bar state where the errors cancel out. The resulting uncorrected matrix error is (see Methods):

$$\mathcal{E}_0 = \begin{cases} 2\sqrt{N}\mu & \text{(MZI)} \\ 2\sqrt{2(\log N - 1.423)}\mu & \text{(MZI + X)} \end{cases} \quad (6)$$

Correlated errors (both corrected and uncorrected) are important because they are tightly connected to the operational bandwidth of the mesh, a critical design parameter for machine learning schemes that require broadband operation, e.g., for parallel processing on wavelength-multiplexed data[50–53]. All beamsplitters are dispersive, and this dispersion leads to a correlated wavelength-dependent splitter error, which can usually be expanded to first order $\mu \approx (d\mu/d\lambda)\Delta\lambda$. Two important wavelength-dependent figures of merit are (1) the tuning range, which refers to the range of $\lambda$ over which the mesh can be programmed to a given accuracy, Fig. 7a, c, and (2) the bandwidth, which is related to the number of wavelength channels that can be (simultaneously) processed by the mesh, Fig. 7b, d. The tuning range is limited by the corrected error $\mathcal{E}_c$, while the bandwidth is limited by the uncorrected error $\mathcal{E}_0$, since a mesh cannot simultaneously error-correct at two different wavelengths. Since the MZI+X design reduces both $\mathcal{E}_0$ and $\mathcal{E}_c$, it leads to enhancements in both the bandwidth and

**Table 1 | Approximate tuning range and bandwidth enhancement factors for mesh sizes up to $N = 512$, Eqs. (7), (32) and (33)**

| $N =$ | 16 | 32 | 64 | 128 | 256 | 512 |
|---|---|---|---|---|---|---|
| $F_{TR} =$ | 5.6× | 10× | 18× | 33× | 61× | 114× |
| $F_{BW} =$ | 2.4× | 2.8× | 3.4× | 4.3× | 5.6× | 7.3× |

**Table 2 | Characteristics of the major tunable crossing types**

| | Complexity | | | Features | | |
|---|---|---|---|---|---|---|
| | Passives | Actives | Area | | | |
| MZI | 2 | 2 | 1.0 | S | | |
| S-MZI | 2 | 2 | 0.8 | | | |
| 3-MZI | 3 | 2 | 1.2 | S | | (P) |
| MZI+X | 3 | 2 | 1.2 | S | B | (P) |
| Suzuki | 3 | 3 | 1.5 | S | | P |
| Miller | 4 | 4 | 2.0 | S | | P |

*S* self-configuration, *B* broadband, *(P)* asymptotically perfect, *P* perfect.

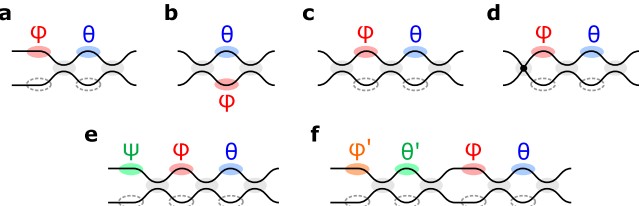

**Fig. 8 | Comparison of crossing types. a** MZI, **b** symmetric (S-MZI)[64], **c** 3-splitter (3-MZI)[32], **d** port-exchanged (MZI+X)[49], **e** Suzuki[24], and **f** Miller[23].

tuning range. The enhancement factors scale as

$$F_{BW} \propto \sqrt{N/\log N}, \quad F_{TR} \propto (N^3/\log N)^{1/4} \qquad (7)$$

and are listed for several mesh sizes in Table 1 (see Methods for details). As Fig. 7c, d illustrates, the MZI+X architecture enjoys a significantly larger tuning range, in addition to modestly greater bandwidth.

Real crossings have a small amount of nonzero crosstalk, quantified by the S-matrix element $S_{21}$; scattering into the forward-facing port leads to a perturbation $R_x(\frac{\pi}{2}) \to R_x(\frac{\pi}{2} + \gamma)$ in the transfer matrix, where $\gamma = 10^{-S_{21}[dB]/20}$. This does not degrade the effectiveness of self-configuration, since the additional scattering angle merely rotates the Riemann sphere Fig. 5c by an additional angle $\gamma \ll 1$, and the forbidden region is still far from $s = 0$. In-plane crossings in silicon can achieve sub-40 dB crosstalk suppression ($\gamma < 0.01$) with insertion losses well below 0.1 dB[54–58]. Unlike directional couplers, crossings are inherently broadband; the insertion loss and crosstalk depend only very weakly on $\lambda$, so any crossing imperfections can be treated as (correctable) wavelength-independent errors that do not affect the bandwidth enhancements of the MZI+crossing scheme. In addition to the forward-scattered light, a 90° crossing will scatter light into the backward-facing port. Back-reflected light can be subsequently reflected in other crossings, leading to a spurious signal that interferes with the forward-propagating light. Provided that the phases of reflected beams are random, these add in quadrature: with amplitude $\gamma^2$ and $O(N^2)$ scattering paths, we expect this to induce an $O(N\gamma^2)$ error, which may be uncorrectable and set a limit on scaling. However, if this effect is small, gradient-based methods or iterative self-configuration may enable correction of these errors.

## Discussion

As photonic circuits grow larger, error tolerance becomes increasingly important. Many techniques exist to manage hardware errors, but all involve a tradeoff between accuracy and complexity. At opposite poles lie "zero-change" error correction, which has limited scalability[16,17,19,59], and "perfect" photonic circuits, which require a larger number of photonic and electronic components[23,24]. This paper has introduced two designs for programmable circuits that strike a tradeoff between these extremes, as shown in Fig. 8 and Table 2, achieving performance that is almost as good as the perfect designs, but with less added complexity (see Supplementary Section 3 for details).

The main insight from this paper is that, by adding a single passive component (either a splitter or a waveguide crossing) to the MZI, we can recover behavior that is asymptotically perfect—that is, the average normalized matrix error *decreases* with size. Our design choices are motivated by the elegant theory of self-configuration by matrix diagonalization[17], where splitting ratios are set to successively zero the off-diagonal elements of the target unitary. By visualizing the MZI state on the Riemann sphere, we can intuitively understand the increased error robustness of our designs in terms of "rotating" the forbidden regions away from the peak probability density. This leads to a several-orders-of-magnitude reduction in post-correction errors compared to the standard MZI mesh. The ability to achieve near-perfect and freely scalable MZI meshes with less complexity than the MZI-doubled designs[23,24] (especially with respect to the number of active components and pads) removes a major roadblock to the realization of very-large-scale nanophotonic systems.

An interesting direction for future work is to explore to what extent multiport interferometers can be made robust to imperfections in the absence of error correction. For example, previous studies of 3-MZI splitters have noted a wavelength-independent coupling ratio for certain parameter choices[32]. Likewise, the near-cancellation of correlated errors in the MZI+crossing architecture explains the $O(\sqrt{N/\log N})$ reduction in the uncorrected error, and corresponding increase in bandwidth. Further design modifications based on the theory of composite pulse sequences[60–62] may allow this imperfect cancellation to be made exact, further improving the bandwidth (and multiplexing capabilities) of linear photonics.

## Methods

### Unitaries and the Riemann sphere

A generic $2 \times 2$ complex-valued matrix has eight degrees of freedom, and a $2 \times 2$ unitary has four. However, the space of $2 \times 2$ unitaries can be divided into equivalence classes based on the splitting ratio $s = T_{11}/T_{12}$. Specifically, any two unitaries are equivalent up to output phases, i.e., $T = \text{diag}(e^{i\psi_1}, e^{i\psi_2})\hat{T}$, if and only if the splitting ratios are the same, $s = \hat{s}$. As a complex number, $s$ can be visualized on the Riemann sphere (Fig. 2d), where the mapping is performed by the stereographic projection $s = (x + iy)/(1 + z)$ (which inverts to $x + iy = 2s/(1 + |s|^2)$, $z = (1 + |s|^2)/(1 - |s|^2)$).

Ordinarily, the distance between matrices is defined as the Frobenius ($L_2$) norm $\|\Delta U\| = (\sum_{mn} |\Delta U_{mn}|^2)^{1/2}$. However, since output phases are corrected in subsequent steps, the most relevant distance metric for a $2 \times 2$ block is the Frobenius norm modulo these phase shifts,

$$d(T, \hat{T}) \equiv \min_\psi \left\| T - \begin{bmatrix} e^{i\psi_1} & \\ & e^{i\psi_2} \end{bmatrix} \hat{T} \right\| = \frac{d(s, \hat{s})}{\sqrt{2}} \qquad (8)$$

where $d(s, \hat{s}) = 2|s - \hat{s}|/\sqrt{(|s|^2 + 1)(|\hat{s}|^2 + 1)}$ is the Euclidean distance between two points on the Riemann sphere.

A common parameterization is $s = e^{i\phi} \tan(\theta/2)$, which represents the splitting ratio of the standard MZI, Fig. 1b. On the Riemann sphere, $(\theta, \phi)$ map to the standard polar coordinates, i.e., $x = \sin(\theta)\cos(\phi)$, $y = \sin(\theta)\sin(\phi)$, $z = \cos(\theta)$.

## Coverage and matrix error derivation

The nulling method relies on successive zeroing of off-diagonal elements to diagonalize the matrix $X$ (initialized to $U$). Each nulling step zeros a single element, increasing the size of the zeroed-out off-diagonal region. Nulling steps are performed in a particular order to ensure that zeroed-out elements remain zero after all subsequent steps[1,2,17]. In a given step, if nulling cannot be achieved perfectly, the "zeroed-out" region of matrix $X$ is left with a residual of magnitude:

$$r = |T_{11}v - T_{12}u| = \sqrt{|u|^2 + |v|^2}\frac{d(s,\hat{s})}{2} \tag{9}$$

where $\hat{s}$ is the target splitting ratio, $s$ is the closest physically realizable value, and $d(s,\hat{s})$ is the Euclidean distance on the Riemann sphere, the same metric used in Eq. (8). The coverage and matrix error depend on (1) the distribution $P(s)$ of target splitting ratios, a function of the distribution of target unitaries, and (2) the locations and sizes of the forbidden regions, a function of the specific mesh implementation (MZI, 3-MZI, MZI+X). For the Haar measure, $P(s)$ depends on an MZI's location in the mesh; for a given $T_{mn}$ it takes the following form[29]:

$$P_{mn}(s) = \frac{n}{4\pi}\left(\frac{z+1}{2}\right)^{n-1} = \frac{n}{4\pi(1+|s|^2)^{n-1}} \tag{10}$$

Here, the density is defined with respect to the area measure on the Riemann sphere

$$d\mu = \sin(\theta)d\theta d\phi = \frac{4}{(1+|s|^2)}d^2s \tag{11}$$

so that $\int P_{mn}(s)d\mu(s) = 1$. Note that, under Eq. (10), $P_{mn}$ is uniform for the lowest row of crossings, and becomes increasingly concentrated as one approaches the triangle's apex; as a result, the overall distribution is strongly biased towards the cross state for large meshes, as shown in Fig. 2e (the same distribution also holds for the rectangular mesh, up to a reordering of the MZIs).

The forbidden regions $\mathcal{F}_\pm$ are centered at opposite poles of the Riemann sphere

$$(s_+, s_-) = \begin{cases} (0, \infty) & \text{(MZI)} \\ (+i, -i) & \text{(3 - MZI)} \\ (\infty, 0) & \text{(MZI + X)} \end{cases} \tag{12}$$

and have radii $R_\pm = 2|\alpha \pm \beta|$. In the case of small hardware errors, where $P(s) \approx P(s_\pm)$ inside each $\mathcal{F}_\pm$, the probability that $\hat{s}$ falls inside the region is given by $\pi R_\pm^2 P(s_\pm)$. The coverage $\mathcal{C}$ is the probability that every $\hat{s}$ avoids the forbidden regions, and is well approximated by

$$\mathcal{C} = \exp\left(-\sum_{mn}\pi(P_{mn}(s_+)\langle R_+^2\rangle + P_{mn}(s_-)\langle R_-^2\rangle)\right) \tag{13}$$

The normalized matrix error $\mathcal{E}_c = \langle|\Delta U|_{\text{rms}}\rangle/\sqrt{N}$ is approximately the quadrature sum of the residuals accumulated during nulling:

$$(\mathcal{E}_c)^2 = \frac{\langle\|\Delta U\|^2\rangle}{N} = \frac{2}{N}\sum_{mn}\langle r_{mn}^2\rangle \tag{14}$$

Here, $\langle\ldots\rangle$ refers to the ensemble average over both Haar-distributed target unitaries $U$[30,31] and the distribution of hardware errors $\alpha, \beta$. We

calculate the mean residual $\langle r^2\rangle$ by averaging Eq. (9) over the distribution $P(s)$. This is simplified in the case of small hardware errors, because the forbidden region is correspondingly small and where we can assume $P(s)$ is approximately constant:

$$\langle r_{mn}^2\rangle = \frac{\pi}{24}\underbrace{\langle|u|^2 + |v|^2\rangle}_{q_{mn}}[P_{mn}(s_+)\langle R_+\rangle^4 + P_{mn}(s_-)\langle R_-\rangle^4] \tag{15}$$

This residual depends on the quantity $q_{mn} = \langle|u|^2 + |v|^2\rangle$, where $(u, v)$ are the highlighted in green in Fig. 2b. Following the Gaussian elimination procedure of a Haar matrix, this evaluates to $q_{mn} = (n+1)/(N+1-m)$.

A detailed description of the nulling algorithm, including a comparison to the local method[19] and global optimization[8–10] (which has a much longer convergence time), is presented in Supplementary Section 1.

## Gaussian errors: MZI and 3-MZI

For the uncorrelated Gaussian perturbation model with $\langle\alpha\rangle_{\text{rms}} = \langle\beta\rangle_{\text{rms}} = \sigma$, the forbidden regions are (statistically) symmetric, with moments $\langle R_\pm^2\rangle = 8\sigma^2$ and $\langle R_\pm^4\rangle = 192\sigma^4$.

For the MZI mesh, the coverage expression Eq. (13) is dominated by the $s = 0$ term, where $P_{mn}(0) = n/4\pi$. Considering only this term, we calculate:

$$\mathcal{C}_{\text{MZI}} = \exp\left(-\frac{\langle R_+^2\rangle}{4}\sum_{mn}n\right) \to e^{-N^3\sigma^2/3} \tag{16}$$

where we have replaced the discrete sum by an integral

$$\sum_{mn}(\ldots) \to \int_0^N \int_0^{N-m}(\ldots)dn\,dm \tag{17}$$

which is valid in the limit of large $N$. Likewise, the top forbidden region dominates the matrix error, so we evaluate Eq. (15) including only the first term in the sum:

$$\langle r_{mn}^2\rangle_{\text{MZI}} \to \frac{n(n+1)}{N+1-m}\frac{\langle R_+^4\rangle}{96} \tag{18}$$

Converting the sum to an integral and substituting $\langle R_+^4\rangle$, we find:

$$(\mathcal{E}_c)_{\text{MZI}} = \sqrt{\frac{N^2}{432}\langle R_+^4\rangle} \to \frac{2}{3}N\sigma^2 \tag{19}$$

Now we redo the calculation for the 3-MZI. In this case, the forbidden regions are located at $s_\pm = \pm i$ and contribute equally to the problem. Following Eq. (13), the coverage is given by:

$$\mathcal{C}_{3-\text{MZI}} = \exp\left(-2\times\sum_{mn}\pi\langle R_\pm^2\rangle P_{mn}(\pm i)\right) \to e^{-16N\sigma^2} \tag{20}$$

Applying Eq. (15), the mean residual left by crossing $T_{mn}$ is:

$$\langle r_{mn}^2\rangle_{3-\text{MZI}} = 2\times\frac{\pi}{24}\underbrace{\frac{n+1}{N+1-m}}_{q_{mn}}\underbrace{\frac{n}{2^{n+1}\pi}}_{P_{mn}(\pm i)}\underbrace{(192\sigma^4)}_{\langle R_+^4\rangle} \tag{21}$$

The factors of two in Eqs. (20)–(26) arise because both forbidden regions contribute equally. This $\langle r_{mn}^2\rangle$ is not slowly-varying with $(m, n)$, so we cannot convert the sums to integrals. We first perform the summation over $n$, which converges rapidly due to the $1/2^{n+1}$ factor (approximating the upper bound to infinity because of the rapid convergence), followed by summation over $m$. We find the

normalized error:

$$(\mathcal{E}_c)_{3-\text{MZI}} = \left( \frac{128\sigma^4}{N} \left[ \sum_{n=1}^{N} \frac{1}{n} - \frac{5}{4} - \log(2) \right] \right)^{1/2} \tag{22}$$

$$\approx 8\sigma^2 \left[ 2 \frac{\log(N) + \gamma_e - \frac{5}{4} - \log(2)}{N} \right]^{1/2}$$

where the discrete sum is approximated using the relation $\sum_{n=1}^{N} n^{-1} \approx \log(N) + \gamma_e$, which defines the Euler–Mascheroni constant $\gamma_e \approx 0.5772$.

### Correlated errors: MZI and MZI+X

Under a correlated error model, $\alpha = \beta = \mu$. In this case, there is only one forbidden region, which for the MZI is centered at $s_+ = 0$, with $R_+ = 4\mu$. The coverage and matrix error for the standard MZI can then be calculated from Eqs. (16) and (19) with the appropriate substitutions for $\langle R_+^2 \rangle$, $\langle R_+^4 \rangle$:

$$\mathcal{C}_{\text{MZI}} = e^{-(2/3)N^3\mu^2} \tag{23}$$

$$(\mathcal{E}_c)_{\text{MZI}} = (4/3^{3/2})N\mu^2 \tag{24}$$

Now consider the MZI+X. The additional crossing rotates the forbidden region to $s_+ \to \infty$. Only the MZIs in the bottom row of the triangle ($n = 1$) contribute to the sums in Eqs. (13) and (14), because the probability distribution Eq. (10) vanishes at $s = \infty$ for the upper rows.

As before, we use the residual formula Eq. (15) to calculate the matrix error. In this case, there is only one forbidden region, centered at $s_+ = \infty$, with $R_+ = 4\mu$. Only the MZIs in the bottom row contribute to the sum, because the probability distribution Eq. (10) vanishes at $s = \infty$ for the upper rows. The coverage is:

$$\mathcal{C}_{\text{MZI}+\text{X}} = \exp\left( -\sum_m \pi \langle R_+^2 \rangle P_{m1}(\infty) \right) \to e^{-4N\mu^2} \tag{25}$$

With the mean residual given by

$$\langle r_{m1}^2 \rangle_{\text{MZI}+\text{X}} = \frac{\pi}{24} \underbrace{\frac{2}{N+1-m}}_{q_{m1}} \underbrace{\frac{1}{4\pi}}_{P_{m1}(\infty)} \underbrace{(256\mu^4)}_{\langle R_+^4 \rangle} \tag{26}$$

and $\langle r_{mn}^2 \rangle = 0$ for $n > 1$, the matrix error evaluates to:

$$(\mathcal{E}_c)_{\text{MZI}+\text{X}} = 4\mu^2 \left[ \frac{2}{3} \frac{\log(N) + \gamma_e - 1}{N} \right]^{1/2} \tag{27}$$

Now we consider the uncorrected matrix error. For the standard MZI mesh, this is $\mathcal{E}_0 = 2\sqrt{N}\mu$[17]. Using the transfer matrix of the standard MZI

$$T_{\alpha,\beta}(\theta,\phi) = R_x\left(\frac{\pi}{4} + \beta\right) \begin{bmatrix} e^{i\theta} & 0 \\ 0 & 1 \end{bmatrix} R_x\left(\frac{\pi}{4} + \alpha\right) \begin{bmatrix} e^{i\phi} & 0 \\ 0 & 1 \end{bmatrix} \tag{28}$$

to first order in $(\alpha, \beta)$, the norm of the matrix error is:

$$|\Delta T|^2_{\text{MZI}} = 2[\cos^2(\theta/2)(\alpha + \beta) + \sin^2(\theta/2)(\alpha - \beta)^2] \tag{29}$$

which is maximized when the MZI is in the cross state $\theta = 0$. For the MZI+crossing (Fig. 5a), we find:

$$T_{\alpha,\beta}^{(\text{X})}(\theta,\phi) = R_x\left(\frac{\pi}{4} + \beta\right) \begin{bmatrix} e^{-i\theta} & 0 \\ 0 & -1 \end{bmatrix} R_x\left(\frac{\pi}{4} + \alpha\right) \begin{bmatrix} e^{-i\phi} & 0 \\ 0 & 1 \end{bmatrix} R_x\left(\frac{\pi}{2}\right) \tag{30}$$

$$= e^{-i(\theta+\phi)} \begin{bmatrix} 1 & 0 \\ 0 & -1 \end{bmatrix} T_{\alpha,-\beta}(\theta,\phi)$$

Up to irrelevant output phases, the effect of the crossing is to flip the relative sign of $\alpha$ and $\beta$, so the component errors appear anti-correlated. As a result, $\|\Delta T\|_{\text{MZI}+\text{X}} \propto \sin(\theta/2)\mu$, which is zero for the cross state. The actual error is found by adding the $\|\Delta T_{mn}\|$ in quadrature and averaging over the probability distribution $P_{mn}(\theta) = n \sin(\theta/2) \cos(\theta/2)^{2n-1}$ (equivalent to Eq. (10)):

$$\mathcal{E}_0 = 2\sqrt{2(\log N + \gamma_e - 2)}\mu \tag{31}$$

For a wavelength-dependent splitter error $\mu \approx (d\mu/d\lambda)\Delta\lambda$, the tuning range and bandwidth can be calculated from the expressions for $\mathcal{E}_c$ (Eq. (27)) and $\mathcal{E}_0$ (Eq. (31)), respectively: the tuning range is the range over which $\mathcal{E}_c(\lambda) < \mathcal{E}_{\max}$, while the bandwidth is the range over which $\mathcal{E}_0(\lambda) < \mathcal{E}_{\max}$:

$$\Delta\lambda_{\text{TR}} = \frac{\sqrt{\mathcal{E}_{\max}}}{|d\lambda/d\mu|} \begin{cases} \frac{3^{3/4}}{\sqrt{N}} & (\text{MZI}) \\ \sqrt{\frac{3N}{2(\log N - 0.42)}} & (\text{MZI} + \text{X}) \end{cases} \tag{32}$$

$$\Delta\lambda_{\text{BW}} = \frac{\mathcal{E}_{\max}}{|d\lambda/d\mu|} \begin{cases} \frac{1}{\sqrt{N}} & (\text{MZI}) \\ \frac{1}{\sqrt{2(\log N - 1.42)}} & (\text{MZI} + \text{X}) \end{cases} \tag{33}$$

From these expressions, we derive the enhancement factors reported in Eqs. (7) and Table 1.

### Neural network model

The optical neural network model is based on the architecture described in ref. 11. Input images are first Fourier transformed, and cropped to a $\sqrt{N} \times \sqrt{N}$ window, where $N$ is the DNN's inner layer size. The signal from this window ($N$ input neurons) passes through two optical layers, with unitary connectivity realized with rectangular meshes. The activation function at the inner layer is realized electro-optically: a fraction of each output field is tapped off and sent to a detector, whose photocurrent modulates the remaining output light[28,39], implementing the activation function:

$$f(E) = \sqrt{1-\alpha} \, e^{-i(g|E|^2 + \phi - \pi)/2} \cos\left(\frac{1}{2}(g|E|^2 + \phi)\right) \tag{34}$$

where $\alpha$ is the power tap fraction, $g$ is the modulator response, and $\phi$ is the phase at zero power. Here, we choose $\alpha = 0.1$, $g = \pi/20$, and $\phi = \pi$, so that $f(E)$ approximates a leaky ReLU in the right power regime. Models of sizes $N = 64$ and $N = 256$ were trained using the NEUROPHOX package[40].

### Simulations and data analysis

All simulations were performed using the MESHES package, an open-source simulator for feedforward photonic circuits that can account for hardware imperfections[63]. Figures 3, 4, 6, and 7 plot multiple instances (usually ≥100) per point; dots show medians while shaded regions show the interquartile range. Source code to produce the plots for this manuscript is provided in the Supplementary material.

## Data availability

All data from this paper can be generated using the MESHES package[63] and source code files provided in the Supplementary material. Source Data are provided with this paper.

## Code availability

Source code files are provided in the Supplementary material.

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

## Acknowledgements

S.B. is supported by an NSF Graduate Research Fellowship. D.E. acknowledges funding from AFOSR (no. FA9550-20-1-0113, FA9550-16-1-0391). The authors thank Prof. David A. B. Miller and Dr Sunil Pai for helpful discussions.

## Author contributions

S.B. and R.H. jointly conceived the idea. R.H. developed the theory, performed the simulations and data analysis, and wrote the manuscript. R.H., S.B., and D.E. contributed to discussion of the results.

## Competing interests

R.H., S.B., and D.E. are inventors on patent applications No. 63/151,103 and 63/196,301 describing methods for self-configuration and error correction in linear photonic circuits.
