## [Peer review file · Nature Communications]

REVIEWER COMMENTS

Reviewer #1 (Remarks to the Author):

The authors presented a theoretical work on the design of multiport unitary optical circuit having both correctability of the fabrication errors and the simplicity. For only correcting the fabrication errors, perfect ways have already been known in literature [22, 23] at the cost of 1.5 - 2× increase of both passives (splitters) and actives (phase shifters). In this paper, the authors proposed two new ways to enable asymptotically perfect error correction (and broader bandwidth for one of the designs) with the cost of 1.5× increase of passives.

The theory clearly described how the reduction of errors is possible by rotating the Riemann sphere. Moreover, the proposed designs have small changes from the standard realization of the unitary optical circuit, making the proposal practically fascinating. Sufficient materials to reproduce the results are provided with the main text. Experimental verifications for the theory would be highly desired especially for MZI+X scheme in which perfectly uncorrelated errors are considered. There are several points that may require reconsiderations and improvements as shown in the following comments for the authors.

1. Some readers may consider that the title, “Infinitely scalable multiport interferometers,” is confusing. Although the proposed designs can correct errors asymptotically perfectly, i.e., it becomes perfect when the port count goes infinite, the device still is not infinitely scalable in the viewpoints of component losses, operating bandwidth, optical or thermal crosstalk and so on. Moreover, as the authors have referred in the main text or in the previous paper from the authors [18], there already have been perfect ways [22, 23] to have the errors corrected, which could also be referred as infinitely scalable.
2. In the introductory paragraphs, clarifications are further required when comparing the proposed scheme to the previous schemes. For example, Table 2 shows that the proposed schemes reduce 33% components and 20% area, which is not clearly shown in the introductory paragraphs. It would also be helpful if the authors could describe other advantages of the proposed schemes as compared to the redundant MZI schemes.
3. The error of the added passive component, denoted by γ in Fig. 1(d), seems not to be described for 3-MZI scheme. It is helpful if the authors can describe the effect of this error for the 3-MZI scheme.
4. In page 3, the authors referred that the 3-MZI structure can have wavelength-independent ratio with certain parameter choices. However, it seems that only limited parameter choices enable wavelength-independent responses in [27]. Is it possible for the proposed method to have broader bandwidth? Intuitively, adding a splitter to the system is seemingly detrimental in the viewpoint of wavelength dependency.
5. The MZI+X scheme described in the section “Broadband Mesh for Correlated Errors” is effective on perfectly correlated errors, i.e., $\alpha - \beta = 0$. However, because realistic errors have finite dispersion

around their average values, there should be small uncorrelated components although the average $\langle \alpha \rangle - \langle \beta \rangle$ can be zero. It would be helpful if the authors can comment on this.

6. In the section of “Neural Network Model,” page 9, the sentence “a fraction of each output field is powers a detector,” needs adjustment.

7. In the final line of the section “Correlated Errors: MZI & MZI+X,” page 9, “Eqs. (7)” needs adjustment.

Reviewer #2 (Remarks to the Author):

The Authors present two new designs for scalable and robust-to-errors Mach-Zender interferometer meshes. Their timely manuscript is relevant to the developments in the past few years in optical and neuromorphic computing. The work is very interesting, and the manuscript is very well written. While their arguments appear to be sound, there are some issues that need to be addressed.

1. Fig. 1 (d, e) shows a cross sign followed by an arrow. The reader may initially interpret the crosses as some form of input. Replacing the arrow with an = sign could be more intuitive. It could also be helpful to guide the reader that they are the same crosses in Fig. 1 (a).

2. Is the “forbidden regions” in MZI mesh the result of the nulling method? If one uses another method, such as global optimization for error correction, would they still exist? It could also be useful to comment on how the nulling method would be realized experimentally.

3. In the comments, it is mentioned that Fig. 2 (e) shows the peak of the probability distribution that occurs in the forbidden regions. However, why the peak occurs as we increase N is not discussed. Also, there is no clear definition of probability density.

4. The top figure of Fig.7 does not explain the orange line in the caption or in the label. The bottom figure of Fig.7 did not explain the lighter red lines. Also, the lower figure has the same BW lines as the top figure, shown in the gray dashed lines. We suggest either erasing the gray lines in the bottom figure or removing the upper figure. Also, matrix error is shown as epsilon. Is it corrected or uncorrected (normalized)? This should also be clear in the captions of the figures. There are too many lines for the bandwidth. It could be simplified by showing fewer lines. Also, the comment in Fig. 7, shortly mentions the tuning range and bandwidth difference. However, there is no further comment on what they mean.

5. The codes given in the supplementary material throw an assertion error due to this line:

```
x = False; assert x.
```

The purpose of this line is not clear and it prevents running the codes. After removing this line data.py runs for two iterations between 0-99, then returns the following error:

```
Invalid use of Boundfunction(array.reshape for array(complex128, 1d, C)) with parameters (Tuple(int32, Literal[int](1)))
```

```
...
```

```
During: resolving callee type: Boundfunction(array.reshape for array(complex128, 1d, C))
```

```
...
```

```
File "meshes\mesh.py", line 40:
```

```
def meshdot_helper(...):
```

```
<source elided>
```

```
...
```

```
T00 = T[0,0,i1:i2].reshape((l,1)); T01 = T[i,j,i1:i2].reshape((l,1))
```

```
^
```

We recommend re-checking the supplementary code with the latest versions of packages and updating the readme file indicating the compatible versions.

Re: NCOMMS-22-19510A
Response to Review Requests

Brief Summary of Changes:

Thank you for taking the time to review this paper. We have incorporated these suggestions into our revised manuscript, which at a high level, included the following changes:

- Added supplementary material to describe the nulling method in detail and compare it to alternative methods, including global optimization (Supp. Sec. 1).
- Analyzed the case of the MZI+X in the presence of imperfect correlation of errors ($\alpha \neq \beta$), informed by wafer-scale variation statistics and our own measurements from calibration of MZIs in meshes. Our analysis reveals that such errors do not pose a substantial problem for meshes at least up to size $N = 512$ (Supp. Sec. 2).
- Clarified the Riemann sphere projection and definition of probability density (Methods).
- Changed Fig. 7 to make the results more clear and intuitive.
- Updated supplementary code and fixed any bugs.

Specific responses to the reviewer comments are given below, with our answers written in blue. Changes to the main text have also been highlighted in the provided PDF.

The Authors

Reviewer #1 (Remarks to the Author):

The authors presented a theoretical work on the design of a multiport unitary optical circuit having both correctability of the fabrication errors and the simplicity. For only correcting the fabrication errors, perfect ways have already been known in literature [22, 23] at the cost of 1.5 - 2 \times increase of both passives (splitters) and actives (phase shifters). In this paper, the authors proposed two new ways to enable asymptotically perfect error correction (and broader bandwidth for one of the designs) with the cost of 1.5 \times increase of passives.

The theory clearly described how the reduction of errors is possible by rotating the Riemann sphere. Moreover, the proposed designs have small changes from the standard realization of the unitary optical circuit, making the proposal practically fascinating. Sufficient materials to reproduce the results are provided with the main text. Experimental verifications for the theory would be highly desired especially for MZI+X scheme in which perfectly uncorrelated errors are considered.

Thank you for the constructive comments. We are also of the opinion that system-level experimental verification is the ultimate test of these claims. At present, we do not have full *system-scale* experiments because the advantages of the paper's proposed schemes are only felt for very large-scale meshes ($N \geq 32$), a regime of great interest to industry, but very difficult for academics due to the both foundry costs and the complexity of PIC design, packaging, and electronic-photonic co-integration at such a large scale (most academic studies are on smaller 4x4 or 8x8 meshes, where the advantages of the 3-MZI and MZI+X are much less pronounced). For this paper, we hope that the analytic and numerical analyses are sufficient; however, as discussed in Question #5 (see below), we can resolve the issue of correlated errors in the MZI+X scheme using a combination of device-level characterization of MZIs on smaller meshes, a survey of published MZI data, and modeling of imperfect error correlations.

There are several points that may require reconsiderations and improvements as shown in the following comments for the authors.

1. Some readers may consider that the title, "Infinitely scalable multiport interferometers," is confusing. Although the proposed designs can correct errors asymptotically perfectly, i.e., it becomes perfect when the port count goes infinite, the device still is not infinitely scalable in the viewpoints of component losses, operating bandwidth, optical or thermal crosstalk and so on. Moreover, as the authors have referred in the main text or in the previous paper from the authors [18], there already have been perfect ways [22, 23] to have the errors corrected, which could also be referred as infinitely scalable.

We agree that the title is perhaps a bit too bombastic, and have changed it to "*Asymptotically Fault-Tolerant Programmable Photonics*", which we hope is more specific and less likely to confuse readers. This title highlights the important and novel property of the meshes (error-correction through self-configuration, which is asymptotically perfect as $N \rightarrow \infty$), so that their scalability, while still finite due to loss, etc., is no longer limited by hardware errors.

2. In the introductory paragraphs, clarifications are further required when comparing the proposed scheme to the previous schemes. For example, Table 2 shows that the proposed schemes reduce 33% components and 20% area, which is not clearly shown in the introductory paragraphs. It would also be helpful if the authors could describe other advantages of the proposed schemes as compared to the redundant MZI schemes.

To better highlight the advantages over previous schemes, we made the following modifications:

- Added the following to the introduction, commenting on MZI-doubled schemes:

The resulting effects on chip area (particularly on emerging high-speed platforms where phase shifters have a large footprint [26, 27]), waveguide length (which affects insertion loss and latency [28]), and electronic complexity (number of pads, traces, DACs / drivers, etc.) make this option unappealing.

- Created a new section in the Supplementary Material (Sec. 3, also referred to in Discussion) focused on length and area estimates based on published works on a range of photonic mesh platforms. An important insight from the published literature is that for many platforms (especially ones supporting rapid modulation, e.g. piezo-optomechanical or Pockels phase shifters), the phase shifters are by far the largest structures in a device, so the reduction in area can be as much as 33% or 50% vs. the Suzuki and Miller schemes, respectively.

3. The error of the added passive component, denoted by γ in Fig. 1(d), seems not to be described for 3-MZI scheme. It is helpful if the authors can describe the effect of this error for the 3-MZI scheme.

Our numerical simulations include errors in all three splitters, so this is properly accounted for and does not affect the error. Errors in the splitting angle of the third splitter will slightly change the location of the forbidden regions because the rotation angle is perturbed, but as long as the errors are small, the regions remain near the equator and the overall argument for the 3-MZI is not affected. We have added comment added to the section “Asymptotically Perfect Photonic Circuits” to this effect:

Small errors γ in the third splitter perturb this rotation angle slightly, but this does not change the structure of the forbidden regions and has little effect on the error correction.

4. In page 3, the authors referred that the 3-MZI structure can have wavelength-independent ratio with certain parameter choices. However, it seems that only limited parameter choices enable wavelength-independent responses in [27]. Is it possible for the proposed method to have broader bandwidth? Intuitively, adding a splitter to the system is seemingly detrimental in the viewpoint of wavelength dependency.

Since the wavelength-independence claim is irrelevant to the results of this paper and may be confusing to readers, we have deleted it. Instead, we bring up the possibility in the concluding paragraph,

For example, previous studies of 3-MZI splitters have noted a wavelength-independent coupling ratio for certain parameter choices.

since this is more of a forward-thinking speculation that goes beyond the current paper; using the wavelength-independent property is one approach worth pursuing to further improve meshes in future designs.

5. The MZI+X scheme described in the section “Broadband Mesh for Correlated Errors” is effective on perfectly correlated errors, i.e., $\alpha - \beta = 0$. However, because realistic errors have finite dispersion around their average values, there should be small uncorrelated components although the average $\langle \alpha \rangle - \langle \beta \rangle$ can be zero. It would be helpful if the authors can comment on this.

This is indeed a valid concern. We have now addressed it in Sec. 2 of the Supplementary Material. The uncorrelated errors ($\alpha - \beta$) are closely related to the cross-port extinction of an MZI, while the correlated term μ is related to the bar-port extinction. It is a well-known fact that in typical MZIs, the cross-port extinction is much larger than the bar-port extinction, and therefore $|\alpha - \beta| \ll \mu$. To verify this, we provide a set of MZI extinction measurements taken on a 3-layer neural network chip containing over 45 MZIs, which show cross-port extinctions about 9 dB higher than in the bar port. This is consistent with a survey of the literature, where extinction ratios are regularly characterized for large-scale MZI-based switches (Table 2, supplementary).

Using these measured extinction ratios, we can create a model of the expected uncorrelated errors, from which we can predict their effect on the error of the MZI+X. This is plotted in Fig. S9. For realistic cross-port extinction of > 35 dB, we find a total matrix error under 2% for meshes as large as $N = 512$. However, it is still true that, in principle, the MZI+X (unlike the 3-MZI) is not asymptotically perfect, so we have added the following comment in the main text:

Ultimately, the scalability of the MZI+X architecture is limited by differential errors $\alpha - \beta$ that arise from local fluctuations in waveguide dimensions. The effect of such errors is analyzed in Supp. Sec. 2. For typical photonic process variations, $|\alpha - \beta| \ll \mu$ and differential errors are insignificant for mesh sizes up to at least $N = 512$.

6. In the section of “Neural Network Model,” page 9, the sentence “a fraction of each output field is powers a detector,” needs adjustment.
7. In the final line of the section “Correlated Errors: MZI & MZI+X,” page 9, “Eqs. (7)” needs adjustment.

We made minor adjustments to typos and grammatical issues in the manuscript. We were not sure about Eqs. (7), which actually lists two equations, and the journal style guidelines are ambiguous, so we will leave this to the copy-editors should the paper be accepted.

Reviewer #2 (Remarks to the Author):

The Authors present two new designs for scalable and robust-to-errors Mach-Zender interferometer meshes. Their timely manuscript is relevant to the developments in the past few years in optical and neuromorphic computing. The work is very interesting, and the manuscript is very well written. While their arguments appear to be sound, there are some issues that need to be addressed.

1. Fig. 1 (d, e) shows a cross sign followed by an arrow. The reader may initially interpret the crosses as some form of input. Replacing the arrow with an = sign could be more intuitive. It could also be helpful to guide the reader that they are the same crosses in Fig. 1 (a).

To clarify that the “X” refers to a crossing, we added arrows connecting it to a crossing in the mesh of Fig. 1(a), which should avoid this confusion. We hesitate to replace the arrow with an = sign because both (d) and (e) show different devices, and saying “X” equal two different structures might confuse the reader. For further clarity, we also added the following to the figure caption: “(d-e) Alternative error-resilient coupler designs proposed in this paper: (d) 3-splitter MZI and (e) MZI+Crossing.”

2. Is the “forbidden regions” in MZI mesh the result of the nulling method? If one uses another method, such as global optimization for error correction, would they still exist? It could also be useful to comment on how the nulling method would be realized experimentally.

The forbidden regions are a very general feature of MZI-based circuits imperfect that limit fidelity regardless of the method, although they appear very explicitly and elegantly when using nulling.

For completeness, and to satisfy our own curiosity about the relative merits of the methods, we have compiled material on the nulling method and its alternatives in Sec. 1 of the Supplementary Information (referenced on p.2 of the main text). This section provides a thorough description of three variants of the nulling method: the original in-silico one used for ideal hardware (papers by Reck, Clements), the measurement-assisted version we recently proposed (published this year in Phys. Rev. Applied), and a refinement to the latter, which is new to this work. We derive analytic expressions for the matrix error under every method, and compare these against numerical data for a number of representative cases (Supplementary Table 1 and Fig. 5). We also compare the nulling method against (1) the local error-correction method of Bandyopadhyay et al., *Optica* 2021, and (2) global optimization based on L-BFGS-B. While global optimization on MZI meshes does slightly better than nulling for Clements meshes (albeit at a great computational cost), the 3-MZI mesh still shows significantly better matrix fidelity. Moreover, nulling and global optimization have the same error scaling with mesh size N , so the scaling advantage of the 3-MZI and MZI+X is preserved.

3. In the comments, it is mentioned that Fig. 2 (e) shows the peak of the probability distribution that occurs in the forbidden regions. However, why the peak occurs as we increase N is not discussed. Also, there is no clear definition of probability density.

This peaking of the probability distribution is a very general phenomenon for meshes (see e.g. Russell, NJP 19, 033007; Pai, PRApp 11, 064044 (2019)), arising from the fact that light must propagate all the way down a mesh’s diagonals to realize generic matrices, meaning that most of the meshes must be very close to the cross state. We added the following to p.3 to clarify this:

This clustering happens because light must propagate all the way down a mesh's diagonals to realize generic unitaries; the forbidden regions disrupt this ballistic transport leading to clipping of off-diagonal matrix elements [10].

To more precisely define the probability density, we added a section “Riemann Sphere” to the Methods section in order to make explicit the mapping between 2x2 unitaries, complex splitting ratios, and the Riemann sphere. The Riemann sphere defines a metric, which is later used (Eqs. (10-11)) of manuscript to define an area measure with respect to which the probability density is defined.

4. The top figure of Fig.7 does not explain the orange line in the caption or in the label. The bottom figure of Fig.7 did not explain the lighter red lines. Also, the lower figure has the same BW lines as the top figure, shown in the gray dashed lines. We suggest either erasing the gray lines in the bottom figure or removing the upper figure. Also, matrix error is shown as epsilon. Is it corrected or uncorrected (normalized)? This should also be clear in the captions of the figures. There are too many lines for the bandwidth. It could be simplified by showing fewer lines. Also, the comment in Fig. 7, shortly mentions the tuning range and bandwidth difference. However, there is no further comment on what they mean.

Thanks for the suggestion. After some thought, we completely remade Fig. 7, separating out the tuning range and bandwidth plots and adding a schematic to illustrate the difference between the regimes where these two quantities apply. We also adjusted the discussion in the main text and added a number of recent references on WDM optical neural networks, which helps motivate the need for broadband meshes.

5. The codes given in the supplementary material throw an assertion error due to this line:

```
<<code stuff here>>
```

We recommend re-checking the supplementary code with the latest versions of packages and updating the readme file indicating the compatible versions.

Wow, great job trying to install the packages and run the code! It is possible that there were some bugs in an earlier version of the Meshes package that was causing this error. We have made some changes to both the code and packages and now see no issues. The code runs when tested on a Mac (2018 15" MBP, OSX 11.6, 6x i7, 16 GB RAM) and on a Linux machine (Ubuntu 16.04.7, 6x i7 + K40 with CUDA 10.1). The Readme file now specifies the compatible version of Meshes as well as the other Python packages to future-proof against changes to any package that could break this code.

REVIEWERS' COMMENTS

Reviewer #1 (Remarks to the Author):

The authors have answered to all questions and made substantial changes to both the manuscript and the supplementary material. I believe that the paper is appropriate for publication.

The followings are small comments regarding the supplementary material.

1. In Supplemental Table 2, please check the type for [28], which is written as “Suzuki” in the main text.
2. Please check the unit for *h* in the Supplemental Table 4, which might be written in [nm]. It would also be helpful for readers if the authors could describe the meanings of each parameter. The “WG”, “Dimensions” groups might be better reconstructed as “Lengths”, “Widths”, respectively?
3. In Supplementary Section 3, the comment “[best ref?]” is left in the text.

Reviewer #2 (Remarks to the Author):

Thanks to the authors for their responses and sound explanations. Therefore I recommend the manuscript be published.

Response to Final Review:

Reviewer #1 (Remarks to the Author):

The authors have answered to all questions and made substantial changes to both the manuscript and the supplementary material. I believe that the paper is appropriate for publication.

The followings are small comments regarding the supplementary material.

1. In Supplemental Table 2, please check the type for [28], which is written as “Suzuki” in the main text.

Corrected.

2. Please check the unit for h in the Supplemental Table 4, which might be written in [nm]. It would also be helpful for readers if the authors could describe the meanings of each parameter. The “WG”, “Dimensions” groups might be better reconstructed as “Lengths”, “Widths”, respectively?

Clarified that w and h are the areal dimensions (for x and y), h is not the waveguide height. Both have dimensions in microns. Defined “WG” in table caption, also highlighted that area equals w*h, to ensure that the reader is not confused into reading h as a vertical dimension.

3. In Supplementary Section 3, the comment “[best ref?]” is left in the text.

Corrected, thank you for catching this.

Reviewer #2 (Remarks to the Author):

Thanks to the authors for their responses and sound explanations. Therefore I recommend the manuscript be published.

No changes were required. We thank the reviewer for their effort in evaluating this manuscript.